# Cutting-Edge Delivery Systems and Adjuvants in Tolerogenic Vaccines: A Review

**DOI:** 10.3390/pharmaceutics14091782

**Published:** 2022-08-25

**Authors:** Chiara Puricelli, Elena Boggio, Casimiro Luca Gigliotti, Ian Stoppa, Salvatore Sutti, Roberta Rolla, Umberto Dianzani

**Affiliations:** Department of Health Sciences, Università del Piemonte Orientale “Amedeo Avogadro”, 28100 Novara, Italy

**Keywords:** tolerance, tolerogenic vaccine, adjuvant, nanoparticles, bystander suppression, dendritic cell, regulatory T cells, immune response, vaccine delivery strategy

## Abstract

Conventional therapies for immune-mediated diseases, including autoimmune disorders, transplant reactions, and allergies, have undergone a radical evolution in the last few decades; however, they are still not specific enough to avoid widespread immunosuppression. The idea that vaccine usage could be extended beyond its traditional immunogenic function by encompassing the ability of vaccines to induce antigen-specific tolerance may revolutionize preventive and therapeutic strategies in several clinical fields that deal with immune-mediated disorders. This approach has been supported by improved data relating to the several mechanisms involved in controlling unwanted immune responses and allowing peripheral tolerance. Given these premises, several approaches have been developed to induce peripheral tolerance against the antigens that are involved in the pathological immune response, including allergens, autoantigens, and alloantigens. Technological innovations, such as nucleic acid manipulation and the advent of micro- and nanoparticles, have further supported these novel preventive and therapeutic approaches. This review focuses on the main strategies used in the development of tolerogenic vaccines, including the technological issues used in their design and the role of “inverse adjuvants”. Even though most studies are still limited to the preclinical field, the enthusiasm generated by their results has prompted some initial clinical trials, and they show great promise for the future management of immune-mediated pathological conditions.

## 1. Introduction

Traditionally, therapeutic approaches to autoimmune diseases have involved the management of symptomatic manifestations with analgesics for pain relief; insulin replacement therapy for autoimmune diabetes; glucocorticoids or interferon for multiple sclerosis (MS); taking standard immunosuppressive drugs such as cyclosporine, mycophenolate mofetil, azathioprine, or tacrolimus; and in the most severe cases, taking intravenous immunoglobulins (IV Ig) and undergoing plasmapheresis. More recently, therapeutic approaches have embraced the use of disease-modifying anti-rheumatic drugs (DMARDs) or biological drugs (such as the anti-tumor necrosis factor, or anti-TNF, the anti-CD20 monoclonal antibody, rituximab, or Janus kinase inhibitors). These “all-purpose” immunosuppressive drugs target the autoreactive process in a more specific manner by modulating inflammation; however, they are still not specific enough to cure the disease, nor do they avoid the broad range of side effects, including increased susceptibility to infections and risk of malignancy in the long term [1,2]. Similarly, in the context of allergic disorders, the most common treatment consists of symptom management with antihistamines, bronchodilators, glucocorticoids, or other immunomodulators, the side effects of which represent an annoying burden for patients and could potentially cause more serious sequelae in the long run [3].

Another perspective may argue that it is worth concentrating on the pathogenic process that causes autoimmune and immune-mediated disorders rather than on its consequences; this involves treating them by inducing tolerance. According to this view, tolerogenic vaccines, also known as “inverse” vaccines, offer a promising opportunity for more specific and efficacious therapies that are able to simultaneously avoid the main drawbacks of immunosuppression; this view is also in accordance with personalized medicine perspectives. In contrast to traditional immunogenic vaccines, which boost the immune response against the antigen of interest and generate immunological memory, tolerogenic vaccines have the opposite effect; they dampen the inflammatory response and induce immune regulation and peripheral tolerance [4]. This approach has already been tried with allergies, where attempts have been made using allergen-specific immunotherapy (AIT) treatments to desensitize the immune system by repeatedly challenging it with incremental doses of the allergen; this makes the immune system tolerant, which can help avoid an inappropriate inflammatory reaction when the immune system is re-challenged with the allergen. Despite being the only treatment with a potential curative capacity, there are still some limitations, including the long duration of the treatments (regimens that last at least three to five-years) and the risk of relapse upon discontinuation [5].

Tolerogenic vaccines also have the potential to be designed using a variety of manufacturing strategies, which may optimize their effectiveness, and they also have the ability to be combined with adjuvants. An adjuvant is any substance or molecule that can enhance the efficacy of the immune response induced by the antigen contained in the vaccine. Moreover, adjuvants also support the function of the vaccine (i.e., whether it is immunogenic or tolerogenic). Immunogenic adjuvants have been used since the introduction of the first inactivated vaccines. Unlike live attenuated vaccines, which confer long-term immunity even after a single dose by eliciting an immune response that is similar to the natural pathogen, inactivated vaccines are less effective, especially in terms of their ability to trigger a cellular response; therefore, adjuvants are very often added to their formulation. They act as both a delivery system, preserving the vaccine after being administered into the injection site and promoting proper initial interactions with local immune cells (the so-called “depot effect”), and as immunopotentiators, given that they provide the necessary additional signals that are required for immune cell activation beyond antigen presentation [4,6,7]. Likewise, adjuvants also have great potential in tolerogenic vaccines, either by downregulating pro-inflammatory responses, defined as “deletional tolerance”, or by enhancing regulatory immune cells, classified by some authors as “dominant tolerance” [7].

This work will present an overview of the main strategies in the design of tolerogenic vaccine platforms for prophylaxis, and the treatment of immune-mediated diseases, with a focus on the main adjuvants used to optimize their efficacy.

## 2. The Concept of Immune Tolerance

The fine regulation of our immune system implies not only its ability to fight against external pathogens and endogenous, potentially dangerous, triggers through innate and adaptive immune responses, but also the ability to keep these processes under control; in other words, preventing these triggers from developing exaggerated and harmful reactions. This includes the concept of tolerance, which could be defined as the ability of the immune system to remain inert when confronted with self-components and innocuous exogenous antigens; this occurs due to a finely tuned educational process which takes place during the earliest development of immune cells. This is particularly meaningful, especially for T cells, which play a pivotal role not only in cell-mediated immunity, but also in the control of B cell-mediated humoral immunity (thymus-dependent antibody responses).

During T cell differentiation in the thymus, thymic epithelial cells and thymic dendritic cells (DCs) modulate a positive and negative selection of autoreactive maturing T cells, inducing apoptosis of high-affinity self-reactive thymocytes or their differentiation into natural (or thymic) Tregs (nTregs or tTregs), a phenomenon known as central tolerance. In particular, developing thymocytes that have an affinity for self-antigens have their fate determined for them. Indeed, strong antigen recognition preferentially moves them towards apoptosis, whereas those that weakly interact with self-antigens are induced to express the transcription factor Forkhead-box-P3 (Foxp3), which is essential for differentiation into natural regulatory T cells (nTregs). This is a process also known as “clonal diversion”, which is due to thymic epithelial cell-derived molecules, such as thymic stromal lymphopoietin (TSLP) and transforming growth factor-β (TGF- β), acting as co-inducers of Foxp3 expression, [8,9].

Conversely, potentially autoreactive maturing B cells in the bone marrow undergo a phenomenon of gene editing to modify their B cell receptor (BCR), thus enabling them to initially recognize self-antigens. In this sense, they are allowed to survive and proceed during the differentiation process; however, they modify their antigenic specificity in order to not be autoreactive [10]. The reason why central tolerance is so fundamental is that the T cell receptor (TCR) and BCR are generated randomly during T and B cell development. This means that some of them will inevitably be directed against self-antigens; thus, natural selection is of paramount importance to prevent this from happening.

Nevertheless, should some cells escape this selection process because of a failure during BCR gene editing, or because some autoantigens are underrepresented in the thymus, their activity would be dampened in the periphery through the induction of anergy. This is due to the absence of co-stimulation during antigen presentation and the promotion of regulatory T cells (Tregs), in this case, peripheral Tregs (pTregs) [11,12]. Interestingly, peripheral tolerance not only aims at inducing functional dormancy towards self-antigens, but also towards exogenous (but harmless) stimuli, against which an immune response would be inappropriate. The best example is represented by microbial or dietary gut-associated antigens, which are tolerated despite not being properly part of the “self” [13].

There are many mechanisms that are fundamental to peripheral tolerance, and they frequently overlap or synergize to potentiate their effects. DCs are key players in this process (Table 1). Although all antigen-presenting cells (APCs) are able to provide an antigenic trigger to T cells, only DCs can determine the final fate of the stimulus (i.e., whether it will be immunogenic or tolerogenic). Indeed, T cell activation requires at least two distinct signals: the first signal involves the interaction between the TCR with the antigenic peptide, which is offered by APCs via the major histocompatibility complex (MHC) molecules, types I and II; the second signal is mediated via the interaction between costimulatory molecules (that are expressed by the T cells) with their ligands (that are expressed by APCs which have been previously activated by inflammatory signals). The best-known type of second signal is mediated by CD28, which is expressed by most T cells, and it interacts with B7.1 (CD80) and B7.2 (CD86), which are expressed via the APC-activated recognition of pathogen-associated molecular patterns (PAMPs), or damage-associated molecular patterns (DAMPs), through their pattern recognition receptors (PRRs). The constraints on these two signals allow the T cells to be activated to only recognize antigens in a “dangerous” context, thus ensuring that a response cannot be triggered by innocuous antigens. Although both signals are fully able to activate T cells, an effective response also requires a third signal, which is mainly mediated by cytokines that are secreted by APCs or other cells within the microenvironment, which causes the activated T cells (specifically T helper cells) to acquire the appropriate effector functions. For instance, IFN-γ, which is produced by NK cells and type I innate lymphoid cells, polarizes T helper cells so that it becomes a Th1 phenotype, which supports cell-mediated immunity (macrophages, NK cells, cytotoxic T lymphocytes) by releasing high levels of IFN-γ and TNFβ. IL-4 that is produced by mast cells and type II innate lymphoid cells drives the differentiation of Th2 cells that produce IL-4, IL-5, and IL-6, which supports B cell-mediated humoral immunity in the absence of IFNγ. Moreover, IL-4, TGF-β, and IL-1 induce Th17 cells that produce IL-17 so that they favor the neutrophil response [7,14].

When DCs are loaded with self- or innocuous non-self-antigens in the absence of PAMPs or DAMPs, they are not able to provide the second signal required for complete T cell activation. As a result, signal three is also dampened, and T cells are not activated despite the antigenic trigger. In other words, they are “tolerized” by the induction of apoptosis (clonal deletion) or anergy (i.e., a state of “functional apathy” in which T cells manage to survive but are unable to synthesize Interleukin (IL)-2, the cytokine responsible for their proliferation and further differentiation).

In the circumstance where an autoreactive T cell also escapes this tolerizing process and is activated by the self-antigen, which is a remote possibility, there is still a chance of keeping it under control, using several mechanisms that act on two levels.

The first level involves “negative” receptors that are expressed by effector T cells several days after their activation, shutting down their activity. These include death receptors, such as Fas, which triggers apoptosis of the senescent effector T cell upon encountering the Fas ligand (FasL), which is widely expressed in the inflamed tissue. Moreover, they include co-inhibitory molecules such as Cytotoxic T-lymphocyte Antigen 4 (CTLA-4) and Programmed Death 1 (PD1), which dampen T cell activity by interacting with their ligands to inhibit progression through the cell cycle and to inhibit the production of IL-2.

The second level involves Tregs, including nTregs which are generated in the thymus, and pTregs which are generated in the peripheral tissues. Moreover, in this case, suboptimal TCR engagement and high levels of TGF-β represent the main triggers for this kind of differentiation. Tregs suppress effector T cell activity through “contact-dependent” regulatory activity (mediated by the Fas, PD-1, and CTLA-4 pathways) by producing inhibitory cytokines such as IL-10 and TGF-β, and by directly killing activated T cells using perforins and granzyme (a phenomenon known as “fratricide”) [9,15]. In addition to T cells, regulatory functions can also be achieved in other cell types, such as B cells and DCs.

The multiple action mechanisms in regulatory cells also imply that there are metabolic pathways which are capable of modulating the immune response. For instance, DCs can express indoleamine 2,3-dioxygenase (IDO), an enzyme that exhibits an antiproliferative and tolerogenic activity [16] with a twofold mechanism. On one hand, it deprives the microenvironment of tryptophan, an essential amino acid used by immune cells as a building block for anabolic reactions, energy metabolism, and proliferation, resulting in metabolic stress [17,18]. The second aspect of this mechanism gives rise to downstream metabolites, collectively called “kynurenines” (Kyn), which can influence the inflammatory and adaptive immune response through the key interaction between Kyn and the aryl hydrocarbon receptor (AhR); this involves a cytosolic transcription factor that translocates to the nucleus upon ligand binding and interacts with the promoters of the target genes. This results in an increased level of anti-inflammatory cytokines, a decreased level of proinflammatory cytokines, a decreased level of Th17, an increased number of Treg cells, and an increased incidence of apoptosis in effector immune cells [16,19]. 

Similarly, the surface receptors and ectoenzymes, CD39 and CD73, which are expressed on a variety of cell types, increase the extracellular production of the immunosuppressive metabolite adenosine, thus shaping the “purinergic halo” surrounding immune cells and modulating their activity [20,21].

Finally, a third metabolic pathway promoting tolerance is mediated by retinoic acid (RA), which is the main metabolite derived from vitamin A after a conversion reaction mediated by the key enzyme, retinaldehyde dehydrogenase (RALD)—mainly RALD2. Most immunological functions of RA are mediated by its interaction with RA nuclear receptors, and interestingly, they mainly take place in the gut-associated lymphoid tissue. In this immunological niche, DCs from the intestinal lamina propria release RA, which promotes T cell homing to the gut and synergizes with TGF-β to induce differentiation into Tregs. Moreover, RA has been shown to downregulate Th1 immune responses while upregulating Th2 activity, thus modulating the immune microenvironment in a tolerogenic direction [22].

The mechanism of tolerance could thus be viewed as a multi-faceted phenomenon; from the very beginning of the process, each phase is carefully controlled and immediately ready to correct a possible failure of the previous step. However, tolerance could be lost in the presence of persistent inflammatory stimuli, the intensity of which is enough to overwhelm such a delicate regulatory process.

## 3. Strategies Used in Tolerogenic Vaccines to Induce Antigen-Specific Tolerance

Improved knowledge concerning tolerance-inducing mechanisms, particularly knowledge that might be considered peripheral, has recently attracted researchers. Moreover, it has paved the way for these naturally occurring processes to be exploited for use in therapeutic or preventive strategies for several autoimmune and immune-mediated disorders which are mainly caused by a loss of tolerance. Most novel immunomodulatory therapeutics include co-inhibitory checkpoint agonists or co-stimulatory checkpoint antagonists that expand the Treg population or downregulate effector T cells; however, there is always a risk involved with such approaches, chiefly their lack of specificity. In other words, unless immunomodulation is limited to the response against a specific antigen or allergen, it might affect several responses, including those that are supposed to eliminate the antigen, thus potentially causing the same side effects as traditional therapies (i.e., widespread immunosuppression). Conversely, using the principle behind conventional immunogenic vaccines, but using it in a tolerogenic manner (triggering immune cells with an autoantigen or allergen while using tolerogenic signals at the same time), could induce antigen-specific tolerance that would not affect other immune responses. Given the variety of tolerogenic pathways, there have been many attempts to mimic them in vitro and in vivo in mice, and although they are still fairly novel, some clinical trials on patients have already been carried out.

### 3.1. Deprivation of Co-Stimulatory Signals

The most likely straightforward approach to induce peripheral tolerance is to deprive immune cells of co-stimulatory signals, as it allows them to become anergic and/or it promotes their conversion into Tregs. This has been achieved with the administration of artificial synthetic APCs, such as nanoparticles (NPs) (see below for a more detailed discussion about nanoparticles in tolerogenic vaccines), which exhibit antigens but lack costimulatory molecules on their surface [23,24]. In this sense, the immune response is not only dampened, but as this downregulation is antigen-specific, it ensures that immune anergy will only take place when that specific antigen is encountered [25] (Figure 1A).

Some promising results have been obtained in preclinical animal models of type 1 diabetes (T1D), MS, and arthritis. For instance, iron oxide nanoparticles coated with antigen-MHC I complexes allowed the suppression of autoreactive CD8^+^ T cells and their acquisition of anergic phenotypes. Similar results were obtained by coating particles with antigen-MHC II conjugates, resulting in the induction of Treg and regulatory B cells. Importantly, the transfer of these cells from vaccinated mice to naïve prediabetic animals conferred protection conferred protection to the development of diabetes, even when they had already experienced the antigen, thus suggesting that tolerance can be transferred and can reverse ongoing responses [23,24].

### 3.2. Inhibition of Pro-Inflammatory Stimuli

As stated above, inflammation provides a bridge between innate and adaptive immune responses as it foreruns the activation of T and B cells, and shapes their subsequent differentiation by providing the signals required for the complete induction of cell-mediated and humoral immunity. In this sense, using the inhibitors of pro-inflammatory mediators as adjuvants, which are co-delivered with antigens, may prove effective at inducing a specific tolerance against the antigens themselves. In a mouse model of inflammatory arthritis, Capini and colleagues showed that an injection of liposomes loaded with lipophilic nuclear factor-κB (NFκB) inhibitors, such as curcumin or quercetin, suppressed effector T cell responses, induced antigen-specific Tregs, and reversed the clinical signs of antigen-induced arthritis [26] (Figure 1B).

Furthermore, a tolerogenic scenario could be achieved by preventing the activity of immune cells at the level of their metabolism and cell cycles. An example of this is the use of the natural molecule rapamycin, derived from *Streptomyces hygroscopicus*, which behaves as an allosteric inhibitor of the mammalian target of the rapamycin (mTOR) complex-1 pathway, which is involved in cell proliferation and differentiation [27]. Combining the adjuvant rapamycin with antigen-containing vehicles, such as polymer particles, can inhibit T cell proliferation and promote Treg expansion, which was demonstrated in an experimental model of MS [28] (Figure 1B). Notably, the generalized immunosuppressive effect that this drug would have if administered alone is now finely tuned to be antigen-specific without inappropriately spreading the tolerogenic effect. 

### 3.3. Harnessing Tolerogenic Physiological Mechanisms

Antigen delivery via the oral route has been demonstrated to be tolerogenic, similarly to the physiological tolerogenic response that is already present in healthy subjects, insofar as exogenous innocuous antigens and intestinal flora colonize the gut [13]. In the intestinal lamina propria and mesenteric lymph nodes, CD103^+^ Tregs are primed by CD103^+^ DCs through the release of TGF-beta and the expression of RALD2and the expression of the enzyme retinaldehyde dehydrogenase type 2 (RALD2), which turns vitamin A into retinoic acid (RA), a potent immunomodulator [29,30].

Allergens are initially orally administered in small doses which are then gradually increased. This is a well-known desensitizing approach that has been widely used to treat IgE- and Th2-mediated food allergies in order to modify the threshold for allergic sensitivity. With oral food challenges (OFC), the immune system becomes “used to the allergen”, and no longer reacts to it, or the immune system’s reaction is at least dampened; this allows the subject to enter a period of sustained unresponsiveness [31]. Harnessing a tolerogenic response towards oral antigens is a strategy that has also been applied to autoimmune disorders. Nevertheless, although oral tolerance appeared to be very effective in preclinical models, the results from human trials are still being investigated, and deserve further study [32]. For example, daily oral administration of bovine myelin in patients with relapsing–remitting (RR) MS reduced the frequency of myelin basic protein (MBP)-specific T cells in a phase I trial [33]; however, it did not improve clinical manifestations of MS in a larger phase III trial [34], and there are also some safety concerns regarding hypersensitive reactions [35]. Moreover, this strategy often requires long-lasting and repeated treatments since naked peptides are rapidly cleared and they only produce transient effects. Consequently, adjuvants would be required to improve their stability, bioavailability, and half-life [32].

As will be detailed in the following sections, nanoparticle-delivered antigens or antigen-coding DNA offer new hope; this is because coupling with NPs offers the advantage of providing an adjuvant to boost a specific tolerogenic response. Collagen-induced arthritis (CIA) was prevented in mice by oral administration using polylactic-co-glycolic acid (PLGA) particles loaded with collagen II (CII) 14 days before immunization with CII [36], whereas oral chitosan nanoparticles containing DNA coding for coagulation factor VIII (FVIII) were effective at inducing a sustained FVIII activity in the absence of neutralizing anti-FVIII antibodies in hemophilic mice [37].

Furthermore, other physiological processes can be mimicked to induce tolerance. For instance, since apoptotic cells generally induce a tolerogenic response, allowing a so-called “neat death” and preventing an inappropriate reaction against dying self-cells, harnessing apoptosis has been proposed for use in a mimicking approach when developing tolerogenic vaccine platforms [38]. This could be achieved by coupling antigens to splenocytes treated with ethylene carbodiimide (ECDI); this induces apoptosis, thus forming apoptotic cell–antigen conjugates that are able to induce antigen-specific tolerance [38]. Alternatively, the high turnover of red blood cells (RBCs), or eryptosis, can be exploited to allow antigens to strongly bind to RBCs; they are then processed in a tolerogenic manner by splenic T cells [39,40]. Moreover, using surrogate APCs, such as liposomes, displaying phosphatidylserine (PS) on their surface allows them to deceive macrophage PS-specific scavenger receptors and induce a tolerogenic phenotype; this includes an increase in anti-inflammatory IL-10 and TGF-β production and reduced pro-inflammatory NFκB and TNF-α signaling (Figure 1D). This model has been exploited to prevent the formation of inhibitory anti-FVIII antibodies in FVIII-treated hemophilic mice [41]. Promising results have also been obtained in mouse models of T1D [42] and experimental acute encephalomyelitis (EAE), which is the animal model of MS [43].

### 3.4. Induction of a Tolerogenic Phenotype

In addition to the imitation of naturally occurring tolerogenic mechanisms, it is also possible to shape the immune microenvironment through the co-delivery of antigens and anti-inflammatory cytokines, such as IL-10, or by engaging tolerogenic receptors. For instance, the tryptophan metabolite Kyn, which is involved in the activation of the tolerogenic Kyn–aryl hydrocarbon receptor (AhR) axis, has been used as an adjuvant. This is coupled to a phage vaccine expressing glutamic acid decarboxylase-65 (GAD65), one of the main autoantigens in T1D, and it has proven to be effective in the prevention of T1D in mouse models [44]. Similar encouraging results were obtained using latex beads coupled with class I MHC molecules and an anti-Fas monoclonal antibody mediating programmed cell death in a murine model of alloskin transplantation [45] (Figure 1C). Moreover, Macauley and colleagues have managed to obtain FVIII-specific tolerance in hemophilia mouse models by vaccinating them with liposomes carrying both FVIII and ligands of CD22 that inhibit the signaling of the BCR, thus dampening the humoral immune response. This tolerogenic vaccine prevented the formation of inhibitory antibodies to FVIII [46]. Of note, this approach is also compatible with the use of protein antigens because the CD22-mediated inhibition is sufficient to overwhelm their intrinsic immunogenicity [25] (Figure 1C). In general, the advantage of tolerance induction over the easier method of depriving the microenvironment of co-stimulatory signals is that, in this way, it is possible to force the immune microenvironment into a tolerogenic phenotype, even in the presence of strong pro-inflammatory stimuli [47].

### 3.5. Dendritic Cell-Based Vaccines

A key function in the complex interplay between innate and adaptive immunity is performed by APCs, whose prototype is represented by DCs. Indeed, DCs can be defined as a cellular bridge linking innate and adaptive immunity. On one hand, they can “sense the dangerous flavor” of either pathogens or cell damage by recognizing PAMPs and DAMPs through their PRRs. On the other hand, they possess the ability to process and display antigens that are loaded on the they possess the ability to process and diplay antigens to the TCR of T cells [48]. Upon stimulation, a maturation program is triggered, which includes the activation of the NF-κB or mTOR intracellular signaling pathways, culminating in the regulation of gene expression and the upregulation of all the necessary costimulators that are required for complete T cell activation (such as CD80 and CD86 and other costimulatory molecules) [27,49]. In addition, they can also determine the direction of the immune response by leaving either a pro-inflammatory or a tolerogenic footprint in the surrounding microenvironment through the release of a variety of cytokines [48]. 

Four major subtypes of DCs have been described: myeloid-derived type 1 and type 2 conventional DCs (cDC1s and cDC2s) are involved in the cross-presentation of antigens and CD8^+^ T cells, and the stimulation and polarization of CD4^+^ Th cells, respectively; lymphoid-derived plasmacytoid DCs (pDCs) quickly secrete type 1 interferons (IFN) in response to viral infections; and monocyte-derived DCs (moDCs) are differentiated from monocytes in the context of inflammation [2,50,51].

Although it was originally thought that immature (or resting) DCs induced T cell anergy via suboptimal antigen presentation and insufficient co-stimulation, the original dogma separating tolerogenic immature DCs from immunogenic mature and migratory DCs has been questioned [11]. This is because, in addition to a quantitative difference between MHC II and co-stimulatory molecule expression, a qualitative difference seems to be necessary for the induction of tolerance *versus* a pro-inflammatory response. Indeed, tolerogenic DCs possess a unique transcriptional program, resulting in a specific cytokine signature (TGF-β, IL-10), the release of immunosuppressive molecules such as nitric oxide, retinoic acid, and IDO, and the expression of ligands for inhibitory co-receptors (PD-L1/2, ICOSL, B7-H4, and B7-H3) that are able to induce the differentiation of T cells into Foxp3^+^ Treg [11,32,48]. Some authors have proposed that this qualitative subtype of DCs should be defined as “semi-mature” [52,53,54].

The most recently obtained knowledge concerning DC properties has paved the way for promising approaches in vaccine platform design. If DCs are key determinants in terms of initiating and mediating the adaptive immune response, it follows that they may be exploited and artificially modulated to serve either a tolerogenic or pro-inflammatory function, depending on what it required.

#### 3.5.1. Ex Vivo DC Education

Ex vivo DC differentiation was one of the first attempts to design DC-based tolerogenic therapies. In brief, patient monocytes or their progenitors, which are recognizable from the expression of the hematopoietic cell marker, CD34, are cultured and allowed to be differentiated in a medium containing DC-dampening factors such as vitamin D, dexamethasone, or immunosuppressive cytokines such as IL-10 and TGF-β. Alternatively, they are genetically engineered to downregulate their expression of co-stimulatory molecules (CD80/86) [2,55]. If these growing DCs are also induced so that they encounter the disease-relevant antigen during the differentiation process, they become tolerogenic against a very specific target after re-infusion in the patient [56,57,58]. Similar attempts, which have been reviewed elsewhere [55], also included the expansion and subsequent reinfusion of mesenchymal stromal cells (MSCs) and Tregs. MSCs have a double advantage in that they lack both co-stimulatory molecules and MHC II expression, and they can be obtained from multiple sources, including from lipoaspirates. In allogeneic transplantation, MSCs obtained from the organ donor can be used to make the recipient more tolerant, thus counteracting the development of immune rejection [55,59].

So far, most studies have considered experimental models of autoimmune diseases in mice; although, some human clinical trials have been performed with promising results, mainly in the context of T1D [60], MS [61], Crohn’s disease [62], and rheumatoid arthritis [63,64,65]. Several aspects still need to be perfected, including optimal delivery route, whether to use parenteral or organ-targeting, the best timings for administration over the course of the disease, and the posology. In addition, the standardization of tolerogenic cell manufacturing techniques is still pending. In fact, despite good tolerability, ex vivo tolerogenic cell vaccination has some drawbacks, including the costly and cumbersome manufacturing process which requires controlled sterile conditions. This is coupled with the fact that monocyte-derived DCs are not exactly the same as their counterparts in vivo [2,66]. Moreover, in the field of Treg-based therapies, a possible safety concern involves the phenotypic instability of Tregs; this instability has been described in relation to the Tregs turning into pathogenic Th17 cells after repeated expansions, subsequently losing their tolerogenic potential, and exacerbating the disease [67,68].

#### 3.5.2. In Vivo DC Targeting Strategies

An alternative to bypassing these obstacles is represented by in vivo DC vaccinations, which consist of DC targeting to induce them into acquiring a tolerogenic phenotype, but in an antigen-specific manner. This kind of DC “education” could be accomplished either through glycan–antigen or antibody–antigen conjugates or using nanoparticles as vectors for both the adjuvant and the antigen; this will be detailed in the following sections.

Among the PRRs of DCs, there is a subfamily of receptors bearing a carbohydrate recognition domain (CRD), including C-type lectin receptors (CLRs) and sialic acid-binding immunoglobulin-type lectins (Siglecs), which specifically recognize glycan moieties on host cells, pathogens, or allergens and they behave both as adhesion molecules and endocytic receptors. They may also mediate intracellular signaling pathways that can eventually instruct other immune cells. In particular, CLRs can engage with either immunoreceptor tyrosine-based activation motifs (ITAMs) or immunoreceptor tyrosine-base inhibitory motifs (ITIMs) depending on the ligand; these generate pro- or anti-inflammatory signals, respectively, whereas Siglecs predominantly produce anti-inflammatory signals using ITIMs or ITIM-like motifs [50]. 

Among some of the first studies on experimental models of autoimmune diseases, the utility of targeting the mannose receptor, DEC205, and mannose-, fucose-, and the *n*-acetylglucosamine-recognizing transmembrane protein, langerin (CD207), has been underlined. It seems that targeting these endocytic receptors on DCs using antigen–anti-DEC205 or antigen–anti-langerin conjugates promotes efficient uptake and presents antigens via the MHC I and MHC II pathways to CD8^+^ and CD4^+^ T cells, respectively. When applied to steady-state DCs, this approach leads to tolerance via different mechanisms, including dominant tolerance via the induction of Tregs, and passive tolerance via the induction of autoreactive T cell anergy and apoptosis [32,69,70]. Interestingly, langerin and DEC205 are often co-expressed, implying that it may be possible to exploit a dual targeting mechanism for Treg induction [71]. To list some examples, the injection of the EAE autoantigen, myelin oligodendrocyte glycoprotein (MOG), which was fused to anti-DEC205-specific antibodies, enhanced the antigen presentation via MHC II using steady-state DCs, which were induced to release IL-10 and TGF-β; this resulted in protection from induced EAE in 90% of the mice treated, compared with none in the control groups [72]. EAE symptom lessening was also obtained after targeting MOG with murine skin and lung DCs after conjugating it with anti-langerin antibodies; this always occurs via the induction of Foxp3^+^ Tregs [73]. Similar results were obtained in experimental models of inflammatory bowel disease (IBD) [74], autoimmune diabetes [75], and arthritis [76]. Unfortunately, data from human trials are yet to be obtained. The main obstacle derives from a differential and much broader expression pattern of DEC205 in human cells compared with murine cells [77,78,79], thus implying that there is a risk of offsite targeting.

DC carbohydrate receptor targeting can also be accomplished by using their natural carbohydrate ligands and coupling them with antigens. This is the case of DC-SIGN and mannose receptors (MRs), which are members of the CLR family that recognize mannose and fucose on many antigens. In particular, DC-SIGN is found only on immature DCs, and its targeting through fucosylated ligands prompts a Th2-biased anti-inflammatory response, Treg expansion, and inhibition of Th1/Th17 immunity [80,81]. 

Conversely, MR expression has been described on murine moDCs, macrophages, and CD1a^+^ dermal DC, and it seems capable of inducing an anti-inflammatory response through IL-10 secretion and PD-L1-mediated apoptosis of autoreactive T cells; this skews the immune response towards an increased Treg/Th1 ratio. Indeed, treatment with the soluble mannosylated proteolipid protein, M-PLP_139–151_, was shown to reduce both the incidence and severity of MS in a rodent model [82]. Vaccination with an epitope of a *Leishmania* analog, however, derived from the receptors of an activated C kinase (LACK), inhibited joint inflammation in an experimental model of autoimmune arthritis [83]. Similarly, encouraging results have been obtained in allergic disease settings using allergoid-mannan conjugates [47,84]; these results included data from clinical trials in humans (EudraCT numbers 2014-005471-88, 2015-000820-27, 2018-002522-23, and 2020-004126-32). 

There is also compelling evidence that allergoid–mannan conjugates could drive the differentiation process of monocytes into tolerogenic or immunogenic moDCs through metabolic reprogramming and epigenetic modulation. A Spanish research group has recently explained that monocyte differentiation from nonatopic and allergic subjects, in the presence of grass pollen–mannan conjugates, yields tolerogenic moDCs with a higher expression of RNA in the typical tolerogenic molecules (IDO, PD-L1, IL-10). This occurs even after lipopolysaccharide (LPS) stimulation, together with a typical metabolic profile characterized by a decreased production of lactate, increased mitochondrial mass, and thus, a shift toward oxidative phosphorylation with greater ATP production, both before and after LPS-mediated stimulation. Moreover, chromatin immunoprecipitation (ChIP) analysis has allowed the molecular basis of epigenetic reprogramming to be simplified, as it relies mainly on histone modification, especially for IL-10 and PD-L1 enhancement, or miRNA involvement, particularly for TNF-α downregulation [47].

Among the ITIM-bearing DC receptors, some Siglecs could be triggered as a mechanism to downregulate the immune response in immune-mediated diseases. The broad Siglec family includes a variety of sialic acid-binding receptors that are differentially expressed on the many DC subtypes, and they show a differential affinity for sialic acid on its position on the underlying glycan (e.g., a better affinity with the α2,3 or α2,6 linkages), and it can perform either *trans*-interactions with sialic acid on different cells, or *cis*-interactions with ligands displayed on the same cell. Of note, these properties are particularly important to sustain paracrine and autocrine tolerogenic signaling, especially in steady-state DCs and Tregs, which are highly α2,6-sialylated. Through the ITIM-mediated engagement of the SH1- and SH2 domain-containing tyrosine phosphatases, (SHP1 and SHP2), Siglecs can boost the tolerance-inducing intracellular signaling pathways, which eventually leads to Treg induction and the decrease of pro-inflammatory Th1 and Th17 cell differentiation. Indeed, targeting pDCs via MOG–anti-Siglec H conjugates or sialylated MOG peptides resulted in dampened inflammatory responses in EAE mice [85,86], whereas the subcutaneous inoculation of sialic acid-modified grass pollen proved to be effective in terms of reducing allergic asthma in mice by reducing antigen-specific Th2 responses and eosinophilic accumulation in the airways [87]. Moreover, in this case, most results were confined to animal studies, and considerable effort should be made to translate these promising data to human settings. Notably, the associations between inhibitory signaling motifs allows Siglecs to behave as tolerogenic receptors even in a pro-inflammatory environment [86]. This is in contrast to other DC carbohydrate receptors such as DEC205, DC-SIGN, or langerin, which are tolerogenic only under steady-state conditions. Consequently, the appropriate context and formulation must be considered when designing tolerogenic vaccines. 

### 3.6. Nucleic Acid-Based Tolerogenic Vaccines

An increased degree of complexity is achieved with nucleic acid-based vaccines that consist of DNA or mRNA molecules which encode the desired antigen(s), either alone or combined with immunomodulators. DNA- and RNA-based vaccines are first internalized by local or target cells. Subsequently, they use the cell machinery to translate into protein products, which are eventually post-translationally modified and subjected to traditional antigen presentation [32]. The expression of the gene of interest is controlled and promoted by coupling the coding sequence with a highly active promoter that is usually derived from cytomegalovirus (CMV). Moreover, with mRNA-based vaccines, antigen expression can be further enhanced by manipulating the mRNA molecule to include additional replicative sequences, most commonly from positive-stranded mRNA viruses, such as alphaviruses, which mediate mRNA auto-amplification and not only simple translation, thus obtaining a self-amplifying (SAM) mRNA vaccine [88]. 

Nucleic acid-based vaccines can either be administered naked, or packaged into microparticles or liposomes. This latter approach seems to improve their uptake and direct them toward the target site. The anatomical location of antigen recognition is of paramount importance since it can influence tolerance. Indeed, tolerogenic nucleic acid-based vaccines are often introduced in immunologically quiescent sites, such as the muscle, or in sites where Treg responses can be easily induced, such as the skin and the liver [32]. Furthermore, an additional step that is part of the manufacturing process of nucleic acid-based vaccines is the removal of intrinsic immunostimulatory components in the nucleotide sequences. For instance, extracellular and double-stranded RNA are inherently pro-inflammatory; therefore, efforts are made to remove double-stranded RNA contaminants to abrogate Toll-like receptor (TLR)-7 activation [89]. Similarly, to prevent TLR-3, TLR-7, and TLR-8 stimulation, uridine is replaced with 1-methylpseudouridine [90,91]. Additionally, in DNA, the number of immunostimulatory CpG motifs is reduced in order to limit TLR-9 activation, whereas immunoinhibitory GpG motifs are increased [92]. 

Intramuscular vaccination of EAE mice with a DNA vaccine encoding a MOG induced MOG-specific Treg expansion reduced the synthesis of IFN-γ, IL-17, and IL-4 after re-stimulation with MOG, thus reducing the clinical and histopathological signs of EAE [93]. The success obtained from the EAE results led to some phase I and phase II clinical trials that tested a DNA-based tolerogenic vaccine encoding MBP, which was found to be safe and able to decrease the number of central nervous system lesions in patients with RR–MS, although, the number was not high enough to be of statistical significance [92]. 

In addition to the antigen, nucleic acid-based vaccines can be engineered to encode immunomodulatory molecules as well, including IL-10, TGF-β, or IL-4, as adjuvants to ensure a tolerogenic response [32]. As an example, Schif-Zuck and colleagues managed to suppress MBP-induced EAE in rats that received separate plasmids encoding MBP_68–86_ or IL-10 under a CMV promoter, either prophylactically or therapeutically [94] (Figure 1E).

Nucleic acid-based vaccines have shown great promise in the prevention and management of immune-mediated disorders, even though the difficult control of the dose, the expression of the kinetics, and the instability of the mRNA molecule have hampered the enthusiasm generated by these methods, as their safety and efficacy remain controversial [92,94,95]. Nevertheless, the recent introduction of mRNA vaccines to prevent and alleviate the severity of the SARS-CoV-2 infection, and their efficacy in combating the COVID-19 pandemic, has broadened the application of nucleic acid-based vaccines. Hopefully, this experience could represent a good starting point in terms of improving knowledge and increasing the prophylactic and therapeutic use of tolerogenic vaccinations that are based on nucleic acids. 

## 4. Epitope Spreading: Hurdle or Advantage?

One of the most concerning issues that may bring tolerogenic vaccines into question is the fear of epitope spreading, a natural phenomenon consisting of unstable self-antigen patterns, which can change over the course of a disease; this jeopardizes the efficacy of these preventive or therapeutic strategies [2,96]. This occurrence is not infrequent in chronic autoimmune disorders, whose pathogenesis may start with an initial antigenic trigger that soon makes way for other antigens that differ only in terms of a single epitope; however, this is enough to cause an autoimmune response, thus prompting a relapse.

A possible solution could be the use of “antigenic cocktails” when designing tolerogenic vaccines; this would induce tolerance in the antigens that are most likely to be involved in disease pathogenesis and epitope spreading. For instance, Smith et al. employed fixed syngeneic splenic APCs that were coupled with a pool of the four most encephalitogenic epitopes to immunize animal models of EAE. The research group was able to prevent the initiation of the active disease that was induced with each peptide (prophylactic tolerogenic vaccination), ameliorate clinical signs, and avoid relapses caused by epitope spreading when the cocktail was administered at the peak of acute disease (therapeutic tolerogenic vaccination) [96]. Likewise, the rationale for using nucleic acid-based vaccines in autoimmune disorders is an attempt to reduce epitope spreading. Employing nucleic acids that are capable of a sustained expression of antigens and immunomodulators can provide a constant and long-lasting tolerogenic stimulus; however, all these practices imply that greater efforts need to be made in the vaccine manufacturing industry, and a more precise knowledge of the epitopes that are most likely to amplify the autoimmune response is required.

A much easier approach would be to exploit the same mechanisms underpinning epitope spreading in order to give tolerogenic vaccines an advantage; in other words, to spread tolerance. Interestingly, Tregs can be activated in an antigen-specific way and then they can expand their immunosuppressive activity beyond their cognate antigen specificity. This phenomenon is known as “bystander suppression”, “linked tolerance”, or “infectious tolerance”, as it looks as though an infectious agent is spreading from one cell to another and influencing the whole microenvironment. An essential requirement is the spatial co-localization of different antigens, which must be presented on the same APC. This implies that reactive T cells have been recruited into regulatory T cell subsets to make them tolerogenic independently of the initial stimulus, thus making them capable of spreading tolerance. In brief, if a Treg is tolerogenic towards an antigen, it can influence a reactive T cell that is specifically intended for an unrelated third-party antigen to become a Treg itself, provided that both antigens are displayed on the same APC, and that it interacts with the two cells. Then, this newly differentiated Treg can mediate tolerance in the unrelated antigen, even when the first antigenic stimulus is no longer present [97]. In other words, the initial Treg can re-educate the reactive T cell by inducing it so that it acquires a Treg phenotype; this can be achieved via direct contact-mediated signaling, through the secretion of IL-10 and TGF-β, or by influencing the APC in a tolerogenic direction [32,98] (Figure 2).

Nucleic acid-based vaccines have proven to be capable of bystander suppression as well. Interestingly, some MPB-based DNA and mRNA vaccines decreased T and B cell responses in MBP, PLP, and MOG. Similarly, in the previously mentioned preclinical experiment by Schiff-Zuck’s research group, the tolerogenic vaccine that was based on a single antigen, but also included an immunomodulatory molecule (IL-10 in this case), demonstrated bystander suppression, as it blocked EAE both when it was induced by MBP_68–86_ and when MBP_87–99_ was used as a trigger [94]. This suggests that using a single epitope can induce tolerance towards multiple noncognate antigens by inducing Tregs that are activated in an antigen-specific way, but that subsequently react and influence the surrounding immune cells in an antigen-non-specific way [91,92,93].

## 5. Focus on the Revolutionary Role of Micro- and Nanoparticle-Based Vaccines

The use of particles in the field of vaccine manufacturing had the original purpose of inducing an immune response and immunological memory against a pathogen, or to boost one’s natural immune surveillance against malignant cells as a way to treat cancer. Several vaccines are already on the market to combat infectious diseases, including those based on virus-like particles, such as the vaccine used to combat human papillomavirus (HPV), and the recently introduced nanoparticle-based vaccines used to combat SARS-CoV-2 [99]; many others are currently under development [100]. Interestingly, the mRNA-based vaccines that have been used to fight the COVID-19 pandemic have been inspired by vaccine platforms that were already available in the oncologic field. For instance, a dual therapy combining the anti-PD1 monoclonal antibody, pembrolizumab, with lipid-encapsulated mRNA encoding tumor neoantigens, is now under study for the treatment of melanoma and other solid tumors [100,101].

However, particle-based vaccines have also been proposed to induce tolerance in order to treat allergies and autoimmune disorders through different mechanisms, from simple antigen delivery to a more complex regulation of immune processes. The peculiar properties of synthetic particles transcend their basic function as antigen cargos. Indeed, it has been demonstrated that they possess adjuvant properties themselves.

### 5.1. Protein-Based Nanoparticles

The simplest design of tolerogenic nanoparticles (tNPs) consists of coating them with MHC molecules that exhibit the antigen of interest. In this sense, tNPs behave as true APCs, but they do not present co-stimulatory molecules, thus resulting in the suppression of T cell responses and the biased differentiation into Tregs in an antigen-specific manner [23,24].

An intriguing alternative is offered by solid biodegradable PLGA particles that can trap loads in their polymer network, which allows its slow and controlled release. This can be further finely tuned by changing the lactic-to-glycolic acid ratio [102]. Cappellano et al. tried a vaccine with such a design in EAE mice, which had both prophylactic and therapeutic purposes. A subcutaneous (SC) inverse vaccination with PLGA loaded with MOG and IL-10 resulted in a protective effect, and the disease severity was decreased in both settings. Moreover, it predominantly deviated the immune response towards Tregs rather than completely suppressing the T cell proliferation and function, thus representing a perfect example of dominant tolerance [103].

In addition to solid particles, liposomes represent another attractive approach for inverse vaccination as they are not intrinsically immunogenic; therefore, APCs can acquire tolerogenic activity during endocytosis. In addition to being loaded with a specific antigen, liposomes can be easily coupled with tolerogenic adjuvants, as was demonstrated with phosphatidylserine (PS), which is a marker of apoptosis [42], or with a ligand of the AhR, which leads to the generation of Tregs as part of the immunomodulatory kynurenine pathway [104,105].

### 5.2. Extracellular Vesicles as Particle-Based Vaccines

Finally, attractive candidates for particle-based tolerogenic vaccine platforms can be offered by the cells themselves. Almost all cell types secrete extracellular vesicles (EVs), which are nanoparticles that consist of a small lipidic bilayer that is involved in cell-to-cell communication, modulation of the extracellular environment, and immune regulation. EVs can be released by Tregs to induce tolerance even at distant sites, probably through the delivery of miRNAs, or signaling surface proteins to peripheral cells [106]. Even though scientific attention has focused mainly on their potential in anti-cancer therapies [107] and infectious diseases [108,109,110], EVs certainly represent an appealing resource for future pioneering studies in the field of tolerogenic vaccines.

There are, however, some limitations to be considered. First, a complete knowledge of the physicochemical properties and the distribution of NPs in vivo is yet to be achieved, especially in humans. Second, the long-term effects of their use are still unknown, particularly regarding side effects. One concern is the ability of NPs to aggregate, which could potentially lead to thrombosis.

### 5.3. Technical Considerations in Particulate Vaccine Manufacturing

In a vaccine manufacturing process, whether tolerogenic or immunogenic, there are many technical issues to take into account in order to achieve the optimal combination of characteristics. This is to ensure appropriate delivery to the desired target and quantitative and qualitative efficacy of the response. In this regard, the physicochemical properties of particulate vaccines are a major determinant in orchestrating immune responses.

First, particle size makes a difference in terms of tissue distribution, cellular uptake, and intracellular processing, and an additional influencing variable concerns the route of administration. For instance, upon intradermal injection, fluid drainage from the interstitial space becomes the predominant way of transport so that size is inversely correlated with transport efficiency, with larger (>50–100 nm) particles often requiring active uptake by tissue-resident DCs [111]; however, when it comes to retention in the lymph nodes, large-sized particles are at an advantage due to more efficient phagocytosis. Moreover, size also determines how particles are taken up by the cells they first encounter. Indeed, although microparticles and large liposomes are preferentially phagocytosed, nanoparticles are mostly taken up via pinocytosis, a phenomenon that consists of “sipping” the extracellular fluid and its small contents through cell membrane invaginations and the creation of endosomes [112]. On the other hand, On the other hand, while the endocytosis of multiple nanoparticles is more efficient than the phagocytosis of a single large particle, the latter can deliver a greater number of antigens to the DC via phagocytosed larger-size particles [113].

Together with size, particle shape also makes a difference. In general, the internalization of spherical particles is almost always favorable, whereas the uptake of rod-shaped particles is always suboptimal as it depends on the critical contact angle when approaching the membrane. This statement can be explained by the fact that endocytosis is mediated by actin filaments that form a “cup” beneath the overlying particle; then, an actin ring squeezes the membrane and pinches it off in the form of an endosome. If the contact angle is too large, the cell membrane will simply embrace the particle; however, the endocytic process will be much less efficient [112].

The material composition of synthetic particles is highly customizable, and it influences delivery efficiency as well as compatibility with the antigen or the immunomodulator. Metal nanoparticles are very stable but not biodegradable, and they necessarily require conjugation with the load. In contrast, liposomes are more easily coupled to the antigen of interest due to their fluidity, and they can bear both hydrophobic molecules on the surface and hydrophilic molecules within their aqueous core [4,25].

Interestingly, particle size, shape, and surface charge, as well as rigidity (which in turn depends on factors such as cholesterol content or lipid transition temperatures), are also determinants for the appropriate skewing of immune responses. It seems that rigid particles in the nanometer range predominantly induce DC cross-presentations, and antigens display CD8^+^ T cells on MHC I [114], together with the preferential activation of Th1 responses. On the other hand, less rigid microparticles, with an optimal size of 1–5 µm, are preferentially processed through the endosomal pathway, resulting in antigen loading on MHC II, the presentation of CD4^+^ T cells, and more frequent skewing toward a Th2 phenotype [112]. In contrast, the generally more rapid lysosomal degradation rate might explain the contradictory behavior of liposomes, as their size modulates the direction of the immune response in the exact opposite way [115,116]. Finally, cationic particles typically drive pro-inflammatory responses, whereas negatively charged surfaces are less immunogenic [25].

## 6. Concluding Remarks

The improved knowledge concerning the physiological mechanisms of immune tolerance, and the idea that vaccines might be used to induce not only a reactive, but also a tolerogenic immune response, might revolutionize the approach to the prevention and treatment of immune-mediated disorders. In this context, the advances in the technological manufacturing of vaccine platforms may represent a new “pillar” in terms of supporting the development of new preventive and therapeutic strategies. The animal models used in preclinical studies suggest that the possible clinical applications of tolerogenic vaccines in humans involve many clinical fields (a summary of all the studies on animals and humans cited in this review is presented in Table 2).

In the context of autoimmunity, a tolerogenic approach is certainly promising, and some studies have already been performed in patients, especially in the context of autoimmune diabetes, MS, and rheumatological disorders such as arthritis [63,64,65].

In addition to autoimmune diseases, tolerogenic vaccine platforms might also be applied to disorders that are thought to be immune-mediated, even though a clear pathological reaction against self-antigens has not been recognized. The best example is perhaps IBD, as its complex pathogenesis also implies a loss of tolerance to antigens expressed by otherwise tolerated commensal gut bacteria; this is due to alterations and functional dysregulation of the intestinal barrier [117]. In a context where the delicate balance between intestinal Tregs and T helper cells is disrupted to the advantage of the latter, particularly regarding Th17 cells, tolerogenic vaccines might represent a chance to re-establish a local equilibrium, thus dampening chronic inflammation and cell-mediated immune responses.

Moving from cell-mediated to type I hypersensitivity, allergies represent another clinical field of potential application. The use of tolerogenic vaccines implies a double benefit, in that it guarantees antigen-specificity on the one hand, thus avoiding the systemic side effects of several drugs being used in the symptomatic treatment of allergic disorders, and it ensures a long-lasting response on the other, especially when the trigger is composed of multiple allergens. Indeed, for a long time, the phenomenon of epitope spreading has been described only in a pathological context, though it can be exploited to spread tolerance beyond the single allergen that is initially presented by the vaccine, thus minimizing the risk of relapses.

Finally, tolerogenic vaccines will likely play a revolutionary role in transplant patients, whose lifelong need for immunosuppressive therapies strongly impacts life quality and raises the risk of opportunist infections and cancer. Inducing specific tolerance only to graft antigens might promote engraftment, limit the need for broader immunosuppression, and guarantee longer-lasting tolerance [118,119].

With regard to the original purpose of vaccines, since their introduction (i.e., their preventive action), the application of tolerogenic vaccine platforms might extend beyond a therapeutic aim to also embrace a prophylactic one; for instance, this may be useful for the primary prevention of autoimmune and immune-mediated disorders in subjects who are highly likely to develop them, such as those with a strong family history of a particular disorder.

Nevertheless, some open questions persist, and they are likely to be the reason why most studies that have been performed so far have focused on pre-clinical models. First, most autoantigens and allergens that are responsible for autoimmune diseases and immune-mediated reactions are still unknown. Second, many autoimmune diseases are driven by more than one autoantigen. In the same way, many allergies are triggered by a broad category of allergens, and their relative weight in the disease pathogenesis is often unclear; therefore, it is still uncertain whether induction of tolerance towards a single or a few autoantigens will be sufficient to counteract an autoimmune disease, or if the induction of a wide organ-specific tolerance would be needed. This may discourage the use of tolerogenic vaccines, as there is a fear that the potentially beneficial tolerogenic response might switch and cause a dangerous boost of the pathology.

The evidence that epitope spreading might be exploited to the advantage of tolerogenic vaccines has revived enthusiasm towards these approaches, especially after some studies and clinical trials have proven that the tolerogenic response can spread beyond the one induced by a single antigen.

On the whole, despite still being in its early stages, the attempt to prevent and treat immune-mediated diseases through the induction of tolerance is a promising strategy that is worth undertaking in the future.

## Figures and Tables

**Figure 1 pharmaceutics-14-01782-f001:**
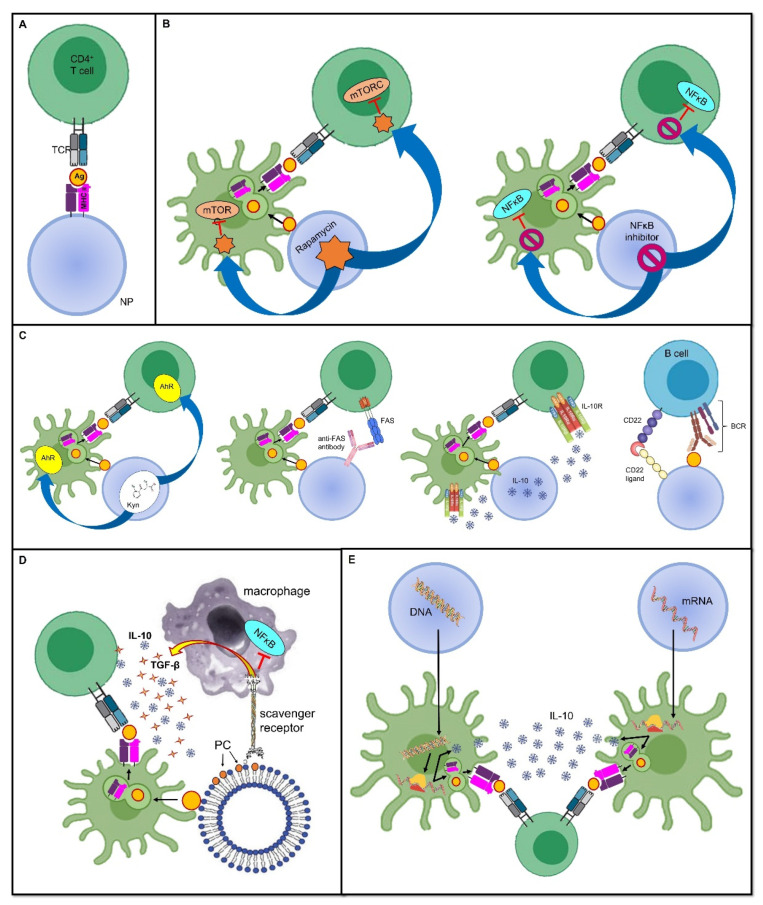
Tolerance-inducing strategies in tolerogenic vaccines. Mechanisms through which tolerogenic vaccines can induce an antigen-specific tolerogenic phenotype during the immune response. For the sake of simplicity, all vaccines are represented as being delivered through nanoparticles, even though several other delivery modes can be employed. Dimensions are not to scale. First, the antigen is usually endocytosed and processed by DCs; thereafter, the antigen is presented on MHC II and naive T cells. The tolerogenic adjuvant is delivered together with the antigen, and it has an effect on both the DCs and T cells interacting with them. (**A**) **Deprivation of co-stimulatory signals.** If the antigen is directly displayed on the MHC II molecule that is being carried by the vaccine, and in the absence of DC mediation and without co-stimulators, the T cell becomes anergic despite antigen recognition. (**B**) **Inhibition of pro-inflammatory stimuli.** Delivering the inhibitors of transcription factors that promote inflammation and cell proliferation is an additional strategy that can dampen the T cell response. (**C**) **Induction of a tolerogenic phenotype.** Engagement between signaling pathways are able to downregulate T or B cell activation, or it can induce programmed cell death. (**D**) **Mimicry of tolerogenic physiological mechanisms.** These strategies harness the physiological “neat death” (apoptosis) of cells; for instance, by displaying the apoptotic marker, PS, thus preventing an inappropriate inflammatory reaction. (**E**) **Nucleic acid-based vaccines.** The nanoparticle-delivered DNA coding for the specific antigen, as well as for a tolerogenic molecule, is first incorporated in the DC nucleus, then transcribed into an mRNA molecule, and eventually translated into a protein antigen plus the tolerogenic molecule. The former is loaded on an MHC II and presented to naive T cells, whereas the latter acts both on the DC itself and on naive T cells to induce a tolerogenic phenotype. In mRNA-based vaccines, the mechanism is very similar, with the only difference being that the transcription step is skipped, and therefore, the mRNA molecule is directly translated due to the DC ribosomes. **Abbreviations**. Ag, antigen; BCR, B cell receptor; CD22, cluster of differentiation 22; IL-10, interleukin-10; mTORC, mammalian target of rapamycin complex; NFκB, nuclear factor-κB; NP, nanoparticle; PC, phosphatidylcholine; TCR, T cell receptor; TGF-β, transforming growth factor-β.

**Figure 2 pharmaceutics-14-01782-f002:**
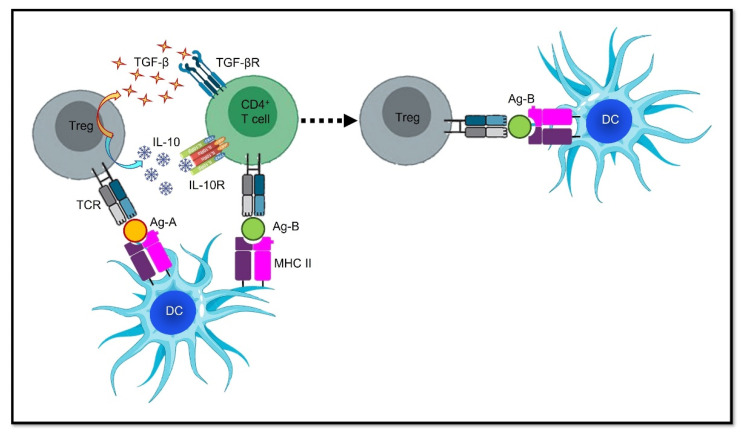
Schematic representation of the phenomenon of epitope spreading. A DC presents two different antigens (Ag-A and Ag-B) to a Treg and an effector CD4+ T cell, respectively. The Treg releases IL-10 and TGF-β, which educates the nearby effector CD4+ T cell so that it becomes a Treg. This results in a downregulation of the immune response towards Ag-B, even though the original tolerogenic stimulus derives from a cell specific Ag-A. **Abbreviations:** Ag, antigen; DC, dendritic cell; IL-10, interleukin-10; IL-10R, interleukin-10 receptor; MHC II, major histocompatibility complex II; TCR, T cell receptor; TGF-β, transforming growth factor-β; TGF-βR, transforming growth factor-β receptor; Treg, regulatory T cell.

**Table 1 pharmaceutics-14-01782-t001:** Mechanisms of peripheral tolerance.

Tolerance Type	Mechanism
**Deletional tolerance: induction of apoptosis (or anergy) of the antigen-specific lymphocytes**	
**Lack of danger signals**	Low expression of MHC I and/or MHC II molecules by APCs → weak signal one. Low expression of co-stimulatory molecules (CD80, CD86, CD28) → weak signal two. Low levels of pro-inflammatory and T cell-polarizing cytokines (IFN-γ, IL-1, IL-2, IL-4, IL-17) → weak signal three.
**Excessive triggering of antigen receptors**	Expression of high levels of autoantigens
**Expression of death receptors**	Tissues expressing high levels of FasL delete activated lymphocytes expressing Fas.
**Dominant tolerance: activity of several types of regulatory (suppressive) cells**	
**Increased production of** **anti-inflammatory cytokines**	Increased synthesis and release of IL-10, TGF-β, IL-4, and NO, result in immune cell anergy and differentiation into several types of regulatory cells.
**Increased expression of** **co-inhibitory molecules**	Increased expression of CTLA-4, PD1, and PD-L1, by both APCs and T cells, mediate a downregulation of T-cell activity despite adequate antigen presentation.
**Direct killing of immune cells (fratricide)**	Perforin- and granzyme-mediated killing of immune cells by several types of cytotoxic lymphocytes.
**Metabolic induction of regulatory T cells**	IDO-mediated conversion of tryptophan into kynurenines, AhR-mediated differentiation into Tregs, and functional suppression of activated T cells. Creation of a “purinergic halo” around immune cells through a CD39- and CD73- mediated increase in immunosuppressive extracellular adenosine. Promotion of Treg differentiation through DC RALD2, turning vitamin A into the immunoregulator, retinoic acid.

**Abbreviations**. Ag, antigen; AhR, aryl hydrocarbon receptor; APC, antigen-presenting cell; BCR, B cell receptor; CTLA-4, Cytotoxic T-lymphocyte Antigen 4; CD, cluster of differentiation; DC, dendritic cell; IDO, indoleamine 2, 3-dioxygenase; IL-10, interleukin-10; MHC, major histocompatibility complex; PD1, Programmed Death 1; PD-L1, Programmed Death Ligand 1; R, Treg, regulatory T cell; RALD, retinaldehyde dehydrogenase.

**Table 2 pharmaceutics-14-01782-t002:** List of the preclinical and clinical studies on tolerogenic vaccines that are cited in this review and a summary of their main characteristics and applications.

Study Group	Year	Study Type *	Animals/Subjects/ Cell Types under Study	Vaccine Type	Clinical Application	Results
**Clemente-Casares et al. [23]**	2016	Preclinical in vivo	NOD mice	Systemic delivery of uncoated nanoparticles or nanoparticles coated with a pMHC that is recognized by the diabetogenic BDC2.5-specific T-cell receptor (TCR).	Animal model of autoimmune diabetes	Expansion of memory-like (CD44^hi^CD62L^low^) FOXP3^−^ T_R_1-like T cells, leading to the suppression of autoantigen-loaded APCs and the differentiation of B cells into autoimmune disease-suppressing B cells.
			Mice expressing a transgenic hybrid MHCII molecule composed of the peptide-binding domain of human HLA-DR4 and the membrane-proximal domain of mouse IE.	Systemic delivery of nanoparticles diplaying mouse collagen (mCII)_259–273_/DR4-IE.	CIA	Expansion of T_R_1-like T cells, reduction of joint inflammation in arthritic mice.
			Mice expressing a transgenic hybrid MHCII molecule composed of the peptide-binding domain of human HLA-DR4 and the membrane-proximal domain of mouse IE.	Systemic delivery of nanoparticles coated with human MOG_97–108_/DR4-IE.	EAE	Systemic expansion of cognate T_R_1-like T cells, EAE blunting.
**Tsai et al. [24]**	2010	Preclinical in vivo	NOD mice	IV injectrion of iron oxide nanoparticles coated with pMHC displaying IGRP.	Animal model of autoimmune diabetes	Expansion of autoregulatory T cells, suppression of local autoantigen presentation by APCs, disease prevention in prediabetic mice, and restoration of glycemic control in diabetic animals.
**Capini et al. [26]**	2009	Preclinical in vivo	Mice primed with Ag to induce inflammatory arthritis	IV or SC injection of egg phosphatidylcholine liposomes loaded with OVA or methylated BSA and a lipophilic NF-κB inhibitor (curcumin, quercetin, or Bay11-7082).	AIA	Suppression of preexisting immune responses in an Ag-specific manner, in situ suppression of APC responsiveness to NF-κB, and induction of Ag-specific FoxP3^+^ regulatory T cells, reduction of joint severity scores.
**Tostanoski et al. [28]**	2016	Preclinical in vivo	Ten to eleven week-old female mice primed with MOG_35–55_ and pertussis toxin to induce EAE	Intranodal injection of polymer particles encapsulating MOG_35–55_ and rapamycin.	EAE	Local LN reorganization, reduced inflammation, systemic expansion of Tregs, reduced T cell infiltration to the CNS, reversal of paralysis after a single treatment.
**Weiner et al. [33]**	1993	Double-blind clinical trial	Thirty subjects with RR–MS	Daily oral administration of capsules of bovime myelin or control protein.	RR–MS	Reduction of the number of T cells that were reactive to MBP in the myelin-treated group, disease exacerbation in 6 out of 15 treated subjects versus 12 out of 15 controls, no toxicity or side effects in etheir group.
**Benson et al. [34]**	1999	Preclinical in vivo	Mice immunized with myelin antigens to induce REAE	Oral administration of high doses of myelin or MBP either before disease induction or during the course of the disease.	REAE	No reduction of in vitro T cell responses, nor protection from disease after oral administration of heterogeneous myelin, versus a reduction in IL-2-, IFN-γ-, and IL-5-secreting MBP-specific T cells and suppression of REAE after repeated oral administration of homogeneous MBP.
**Bielekova et al. [35]**	2000	Phase II clinical trial	Twenty-four patients with RR–MS	Weekly subcutaneous administration of 50 mg of CGP77116, an altered peptide ligand mimicking MBP_83–99_.	RR–MS	Exacerbation of MS in three patients, trial interruption due to poor tolerance of the administered peptide.
**Weiner [34] ****	/	Multicenter clinical trial controlled for patients’ gender and steroid treatments	Five hundred patients with RR–MS	Daily oral administration of 300 mg of bovine myelin or casein.	RR–MS	No clinical improvement after bovine myelin administration. Results not published.
**Kim et al. [36]**	2002	Preclinical in vivo	Mice immunized with CII to induce CIA 14 days after vaccine administration	Prophylactic oral administration of PLGA nanoparticles, entrapping CII 14 days before immunization.	CIA	Higher level of TGF-β mRNA expression in Peyer’s patches, lower level of TNF-α mRNA expression in draining lymph nodes in treated mice, presence of serum anti-CII IgG antibodies and CII-specific T cell proliferation, reduced incidence and severity of CIA.
**Dhadwar et al. [37]**	2010	Preclinical in vivo	FVIII KO mice	Oral administration of chitosan nanoparticles containing canine FVIII DNA on two biconsecutive days.	Animal model of hemophilia	Decrease in aPTT, stable clot formation, restoration of FVIII activity, disappearance of FVIII inhibitors and non-neutralizing anti-FVIII antibodies.
**Kontos et al. [38]**	2012	Preclinical in vivo	NOD mice receiving transgenic diabetogenic CD4+ T cells to induce rapid diabetes onset	IV administration of a peptide antigen fused to an erythrocyte-binding antibody.	Animal model of autoimmune diabetes	Prevention of diabetes onset.
**Yau et al. [41]**	2017	Preclinical in vivo	FVIII KO mice	Six weekly SC injections of FVIII in the absence and presence of liposomes bearing s-PS, followed by a re-challenge with four weekly SC injections of solely FVIII in half of the animals, and OVA in the other half 24 h after liposome treatment.	Animal model of hemophilia	Reduction in both the total anti-FVIII titers and neutralizing titers, robust immune response to the irrelevant antigen OVA, irrespective of treatment, with s-PS liposomes or free FVIII.
**Pujol-Autonell et al. [42]**	2015	Preclinical in vivo	Normoglycemic NOD mice at 8 weeks of age (at least 12 per group)	Single IP dose of 3.5 mg of PS-liposomes (empty or insulin peptide-filled) or a saline solution, followed by monitoring until 30 weeks of age.	Animal model of autoimmune diabetes	Induction of tolerogenic dendritic cells, impairment of antigen-specific autoreactive T cell proliferation, reduction of insulitis severity and prevention of T1D, significant expansion of antigen-specific CD4+ T cells in the spleen, pancreatic, and mediastinal lymph nodes in mice treated with insulin-filled PS-liposomes compared with those receiving empty PS-liposomes or saline.
**Roberts et al. [43]**	2015	Preclinical in vivo	Mice challenged with a SC MOG_35–55_ peptide using a complete Freund’s adjuvant and heat-killed mycobacterium tuberculosis to induce EAE	Four-day-long IV administration of 50 μg of liposomal PS, PS-loaded PLGA nanorods, or blank PLGA nanoparticles seven days after the MOG_35–55_ challenge.	EAE	Significant suppression of IL-6 and IL-12 by DCs and of IFN-γ, IL-2, IL-6, and TNF-α by T cells, decreased disease burden, more pronounced in the mouse group receiving PS-loaded PLGA nanorods versus liposomal PS.
**Sun et al. [44]**	2021	Preclinical in vivo	Female NOD mice at 4–6 weeks of age (12 animals per group)	Double SC immunization with GAD65 phage vaccine alone or mixed with 200 mg of Kyn.	Animal model of autoimmune diabetes	In the group receiving a GAD65 phage vaccine mixed with Kyn, there was an enhancement of Th2-mediated immune responses, regulation of the Th1/Th2 imbalance, increased secretion of Th2 cytokines and of CD4+ CD25+ Foxp3^+^ T cells, suppression of DC maturation and GAD65-specific T lymphocyte proliferation, and prevention of hyperglycemia in 60% of mice for at least one month.
**Shen et al. [45]**	2011	Preclinical in vivo	H-2K(d) mice grafted with skin squares from another mouse strain (H-2K(b))	IV administration of latex beads covalently coupled to H-2K(b)/peptide monomers and anti-Fas mAb after skin engraftment.	Allograft rejection	Prolongation of alloskin graft survival for 6 days, 60% decrease of H-2K(b) antigen-alloreactive T cells.
**Macauley et al. [46]**	2013	Preclinical in vivo	FVIII-deficient mice challenged with human FVIII	Liposomal nanoparticles displaying CD22 ligands (STALs) and FVIII.	Hemophilia mouse model	Prevention of anti-FVIII inhibitory antibody formation, prevention of bleeding in the tail-cut assay after human FVIII administration.
**Benito-Villalvilla et al. [47]**	2022	Preclinical in vitro	Monocytes from nonatopic and allergic human subjects	Culture of monocytes with allergoid–mannan conjugates.	Allergy	Differentiation of monocytes into stable tolerogenic DCs, production of fewer cytokines, a lower TNF-α/IL-10 ratio, and higher expression of the tolerogenic molecules PDL1, IDO, SOCS1, SOCS3, and IL10 after LPS stimulation, induction of higher numbers of functional FOXP3^+^ Tregs, shift of glucose metabolism due to the Warburg effect, lactate production due to mitochondrial oxidative phosphorylation, epigenetic reprogramming within tolerogenic loci, increased expression of the anti-inflammatory miRNA-146a/b, and decreased expression of proinflammatory miRNA-155.
**Ma et al. [56]**	2003	Preclinical in vivo	Female six to seven week-old NOD mice	Single IV injection of MBDCs cultured with ODNs containing a consensus of NF-κB binding sites that inhibit NF-κB activity, stimulated with LPS, and primed with lysate of pancreatic islets for the last 48 h of culture.	Animal model of autoimmune diabetes	Suppression of costimulatory molecule expression, IL-12 production, and immunostimulatory capacity in presenting allo- and islet-associated antigens using NF-κB ODN DC, prevention of diabetes onset, pancreatic T-cell hyporesponsiveness in islet antigens with low production of IFN-γ and IL-2.
**Verginis et al. [57]**	2005	Preclinical in vivo	Female six to eight-week old female mice challenged with Tg	IV administration of TNF-α-treated, semimature BMDCs pulsed with Tg or OVA.	EAT	Inhibition of the subsequent development of Tg-induced EAT after administration of DCs pulsed with Tg, but not with OVA.
**Benham et al. [58]**	2015	Single-center, open-labeled, human phase I trial	Eighteen HLA risk genotype-positive RA patients with citrullinated peptide-specific autoimmunity and 16 RA patients as controls	Single ID administration of Rheumavax: autologous DCs modified with a NF-kB inhibitor that is exposed to four citrullinated peptide antigens.	RA	Reduction of effector T cells and an increased ratio of Treg to effector T cells, reduction in serum IL-15, IL-29, CX3CL1, and CXCL11, and reduced T cell IL-6 responses to vimentin447–455–Cit450 relative to controls, reduction in disease activity scores one month after treatment, good tolerance with minimal side effects.
**Bell et al. [63]**	2017	Unblinded, randomised, controlled, dose escalation phase I trial	Patients with knee RA, three patients per cohort	Arthroscopic injection of 1 × 10^6^, 3 × 10^6^ or 10 × 10^6^ autologous DCs differentiated from PBMCs obtained by leukapheresis and tolerized using dexamethasone and vitamin D.	RA	Stabilization of knee symptoms in two patients receiving 10 × 10^6^ tolDC but no systemic clinical or immunomodulatory effects were detectable.
**Harry et al. [64]**	2010	Preclinical in vitro	Monocyte-derived DCs from RA patients and controls	Culture of monocyte-derived DCs with the immunosuppressive drugs dexamethasone, vitamin D₃, and the immunomodulator monophosphoryl lipid A.	RA	Induction of a tolerogenic phenotype in DCs from RA patients (reduced costimulatory molecules, low production of proinflammatory cytokines, and impaired stimulation of autologous antigen-specific T cells), comparable to healthy control tolDCs, refractoriness to further challenge with proinflammatory mediators.
**Giannoukakis et al. [60]**	2011	Randomized, double-blind, phase I study	Ten insulin-requiring type 1 diabetic patients between 18 and 60 years of age	ID abdominal administration of 10 million autologous BMDCs, unmanipulated or engineered ex vivo toward an immunosuppressive state (with a mixture of antisense oligonucleotides targeting the primary transcripts of CD40, CD80, and CD86) once every two weeks for a total of four administrations.	T1D	Increased frequency of peripheral B220+ CD11c − B cells, but no statistically relevant differences in immune populations or biochemical, hematological, and immune biomarkers compared with baseline results.
**Jauregui-Amezaga et al. [62]**	2015	Phase I, single-centre, sequential-cohorts, dose-range study	Nine refractory-CD patients	Single versus three biweekly IP injection of tolDCs (treated with dexamethasone and vitamin A) at escalating doses (2 × 10^6^/5 × 10^6^/10 × 10^6^), one-year follow-up.	CD	Clinical remission in 11% of participants, clinical response in 22% of participants, and lesion improvement in 33% of patients. An increase of circulating Tregs and decrease in IFN-γ levels. Withdrawal suffered by three patients due to CD symptoms worsening.
**Zubizarreta et al. [61]**	2019	Human phase 1b clinical trial	Twelve patients, 8 with MS and 4 with NMOSD	IV administration of three doses of autologous DCs differentiated from PBMCs obtained by leukapheresis and tolerized with dexamethasone.	MS and NMOSDs	Good tolerance, significant increase in the production of IL-10 and IFN-γ levels, increased frequency of Tr1 cells and switch towards Th2 responses by week 12 of follow-up.
**NCT02354911**	2015–2019	Randomized, quadruple-blind, placebo-controlled, cross-over, phase II clinical trial	24 subjects with T1D	Abdominal ID injection of autologous DCs harvested by leukapheresis and engineered ex vivo via incubation with antisense DNA oligonucleotides that target the primary transcripts of CD40, CD80, and CD86 to convert to active immunoregulators (four separate injections at 2 week intervals).	T1D	Primary endpoint: change from baseline with a mean of 2 h, AUC for plasma C-peptide at 12 and 24 months. Secondary endpoints: adverse events, changes in HbA1c profile, insulin needed, immunological changes.
**NCT02622763**	2015–2019	Randomized, single-blinded, phase I clinical trial	Three patients with refractory CD	Intralesional administration of tolDCs.	CD	Primary endpoints: adverse events, clinical response. Secondary endpoints: clinical remission, quality of life, lesion severity.
**NCT03337165**	2016–2019	Open-label, phase I clinical trial	Ten patients with RA	Single intra-articular injection (into the knee joint) of autologous moDCs generated in the presence of IFN-α/GM-CSF and tolerized with dexamethasone.	RA	Primary endpoint: adverse events. Secondary endpoint: change in clinical severity.
**NCT02903537 (TOLERVIT-MS)**	2017–2021	Non-randomized, open-label, phase I clinical trial	Sixteen participants with RR–MS	Intranodal administration of autologous moDCs tolerized with vitamin-D3 and pulsed with myelin peptides.	MS	Primary endpoints: adverse events, neurologic, and imaging changes. Secondary endpoints: relapse rate, clinical efficacy, immunological changes
**NCT02618902** **(MS-tolDC)**	2017–2021	Open-label, dose-escalating, phase I clinical trial	Nine subjects with MS	ID injection of tolDCs in the subclavicular region.	MS	Primary endpoints: safety and feasibility. Secondary endpoints: clinical impact, lesion severity, immunological changes.
**ONE Study ATDC Trial** **(NCT02252055)**	2015–2018	Phase I/II monocentric trial	Eleven patients with renal insufficiency receiving a first kidney transplant from a living donor	IV administration of autologous tolDCs the day before transplantation.	Organ transplantation	Primary endopint: incidence of biopsy-confirmed acute rejection Secondary endpoints: incidence of short- and long-term complications (including malignancy), immunological conditions, total immunosuppressive burden
**NCT02283671**	2015–2019	Phase I, open-label clinical trial	Twenty patients with MS or NMO	IV administration of tolDCs loaded with myelin peptides every two weeks for a total of three administrations per patient.	MS, NMO	Primary endpoint: tolerability and safety Secondary endopoints: disease severity, changes in immunological profile
**Mahnke et al. [69]**	2003	Preclinical in vivo	Mice challenged with OVA	SC injection of OVA coupled with anti-DEC-205 mAb.	Antigen-induced hypersensitivity reactions	In vivo induction of a tolerogenic phenotype in DCs, suppression of CD4^+^ T-cell-mediated hypersensitivity reactions (reduced IL-2 production), reduction of CD8^+^ T-cell–mediated allergic reactions.
**Ring et al. [72]**	2013	Preclinical in vivo	Mice challenged with MOG to induce EAE	IV injection of scFv specific for DEC-205 fused MOG 7, administered three days before the induction of EAE, or one and four days after the induction of EAE.	EAE	Elevated numbers of highly activated, IL-10–producing CD4 + CD25 + Foxp3+ Tregs, increased levels of TGF-β, protection from EAE or EAE abrogation in 90% of the vaccinated mice, compared with isotype controls and uninjected mice.
**Wadwa et al. [74]**	2016	Preclinical in vivo	VILLIN-HA transgenic mice receiving HA- specific CD4^+^ Foxp3^−^ T cells IV to induce intestinal inflammation	IP injection of antibody–antigen complex consisting of the immunogenic HA_110–120_ peptide coupled to an anti-DEC-205 mAb 2 and one day before adoptive transfer of HA-specific CD4^+^Foxp3^−^ T cells.	IBD	Reduction of intestinal inflammation, conversion of naive HA-specific CD4 + Foxp3 − T cells into HA-specific CD4 + Foxp3 + Tregs.
**Mukhopadhaya et al. [75]**	2008	Preclinical in vivo	NOD mice	Mimotope of a β cell peptide coupled with an anti-DEC-205 mAb.	Animal model of autoimmune diabetes	Deletion of autoreactive CD8+ T cells, absence of immune response after a rechallenge with the mimotope peptide in the adjuvant.
**Spiering et al. [76]**	2015	Preclinical in vivo	Mice receiving IP injection of human PG_70–84_ to induce PGIA	IP injection of human PG_70–84_ coupled with anti-DEC-205 mAb, 10 or 20 days prior to the induction of PGIA.	PGIA	No alterations in the proportion of Foxp3^+^ Treg cells, but reduced numbers of IL-17^+^ and IFN-γ^+^ cells, impaired germinal center formation, and reduced serum levels of PG-specific IgG2a antibodies.
**Yang et al. [83]**	2018	Preclinical in vivo	Mice challenged with collagen to induce CIA	Vaccination or treatment with a LACK_156–173_ epitope expression plasmid or polypeptide.	CIA	Amelioration of arthritis severity, improvement in the balance of effector T cells in synovial tissue towards a Th2 polarization compared with untreated arthritis controls, decreased expression of TLR4 expression, decreased macrophage activation.
**Sirvent et al. [84]**	2016	In vitro and in vivo	hmoDCs from patients with grass pollen allergy	Treatment of hmoDCs with glutaraldehyde-polymerized grass pollen allergoids coupled to nonoxidized mannan (PM), glutaraldehyde-polymerized allergoids (P), native grass pollen extracts (N), or oxidized PM.	Grass pollen allergy	Induction of a tolerogenic phenotype in hmoDCs treated with PM compared with P or N, higher levels of IL-6 and IL-10 and lower IL-4 levels, induction of hmoDC-induced generation of Th1 cells and a Th2/Treg cell shift in favor of Tregs, abolition of these effects after oxidation of PM (due to alteration of the carbohydrate structure).
			Mouse models	Subcutaneous immunization of mice with PM, P, N, or oxidized PM.		Comparing mice immunized with PM to those immunized with P or N. Higher percentage of Tregs in splenocytes, higher ratio of serum IgG2a/IgE levels that are specific to native grass pollen extracts, higher IL-10/IL-4 ratios after in vitro stimulation of mouse splenocytes with the native grass pollen extract, abolition of these effects after oxidation of PM (due to alteration of the carbohydrate structure).
**Loschko et al. [85]**	2011	Preclinical in vivo	Mice challenged with MOG on the same day and two days after immunization	IP injection of OVA coupled with anti-Siglec mAb.	EAE	Reduction of Th1/Th17 cell polarization, but no generation or expansion of MOG-specific Tregs nor deviation to Th2 or Tr1 cells, EAE onset delay and reduction of disease severity.
**Hesse et al. [87]**	2019	Preclinical in vivo	Grass pollen-sensitized mice	SC injection of unmodified or sialylated PhI-p5a peptides before grass pollen challenge.	Airway inflammation due to grass pollen allergy	Increased T-cell activation, enhanced numbers of FoxP3^+^ T cells both in vitro and in vivo, and increased suppression of Th2 cells and eosinophilic inflammation in lung tissue after peptide sialylation compared with unmodified peptides.
**Fissolo et al. [93]**	2012	Preclinical in vivo	Mice immunized with MOG_35–55_ peptide to induce EAE	IM administration of a DNA vaccine encoding MOG prophylactically or therapeutically.	EAE	Reduction of clinical and histopathological signs of EAE in both prophylactic and therapeutic settings, dampening of antigen-specific proinflammatory Th1 and Th17 immune responses, expansion of Tregs in the periphery, upregulation in the CNS of genes encoding neurotrophic factors and proteins involved in remyelination.
**Bar-Or et al. [92]**	2007	Randomized, double-blind, multi-centered, placebo-controlled clinical trial	Thirty patients with RR–MS or SP–MSMS patients	IM injection of a DNA vaccine encoding full-length MBP (BHT-3009) at weeks 1, 3, 5, and 9 at three testing doses (0.5 mg, 1.5 mg, and 3 mg).	MS	Good tolerability and safety, marked decrease in proliferation of IFN-γ-producing CD4^+^ T cells, reduction in titers of myelin-specific autoantibodies in CSF, reduction of inflammatory lesions on brain MRIs.
**Schif-Zuck et al. [94]**	2022	Preclinical in vivo	Seven week-old female mice immunized with SC MBP68–86 (six mice per group)	Treatment with plasmid DNA encoding MBP_68–86_, or plasmid DNA encoding IL-10, or both plasmids coadministered.	EAE	Fast EAE remission only in rats given coadministration of the IL-10-encoding DNA plasmid together with the MBP-encoding plasmids, improvement of EAE-induced CNS lesions at histology (elevation in Ag-specific T cells producing IL-10, increase in apoptosis of cells around high endothelial venules, induction of Tr1-induced active tolerance).
**Garren et al. [95]**	2008	Randomized, placebo-controlled, multi-centered phase II trial	Two hundred and eighty-nine RR–MS patients randomized into three groups (1:1:1)	IM injection of placebo, 0.5 mg BHT-3009 (a DNA vaccine encoding full-length MBP), or 1.5 mg BHT-3009 at weeks 0, 2, 4, and every four weeks thereafter until week 44.	MS	Reduction of the rate of new enhancing MRI lesions after treatment with the lower dose (0.5 mg), lack of efficacy at higher doses.
**Smith et al. [96]**	2006	Preclinical in vivo	Eight to ten-week-old female mice immunized with SC encephalitogenic peptides to induce EAE	IV administration of splenocytes coupled with a peptide cocktail of four distinct encephalitogenic epitopes immediately after immunization or at the peak of acute disease.	EAE	Inhibition of active EAE initiation, prevention of activation of autoreactive Th1 cells and subsequent infiltration of inflammatory cells into the CNS, prevention of clinical relapses due to epitope spreading and increased production of the anti-inflammatory cytokines TGF-β and/or IL-10 in both the periphery and the CNS when administered at the peak of the acute disease.
**Cappellano et al. [103]**	2014	Preclinical in vivo	Four to eight-week old female mice immunized to induce EAE	SC injection of PLGA nanoparticles loaded with MOG 35–55 and recombinant IL-10 either 30 and 15 days before EAE induction or 8 and 22 days after EAE induction.	EAE	Inhibition of EAE development after prophylactic vaccination and significant amelioration of EAE after therapeutic vaccination, decrease in histopathological lesions, reduced secretion of IL-17 and IFN-γ.
**Kenison et al. [105]**	2020	Preclinical in vivo	Eight to ten-week old female mice immunized with SC MOG_35–55_ to induce EAE	Administration of NLPs loaded with an AhR agonist and a T cell epitope from MOG_35–55_; either SC once a week beginning on day one after disease induction, IV once on day seven for disease prevention, or once on day fifteen for disease reversal.	EAE	EAE suppression, expansion of MOG_35–55_-specific FoxP3^+^ Tregs and Tr1 cells, reduction in CNS-infiltrating effector T cells, amelioration of chronic progressive EAE.

**Abbreviations**. Ag, antigen; AhR, aryl hydrocarbon receptor; AIA, antigen-induced inflammatory arthritis; AUC, area under the curve; BAT, basophil activation test; BMDC, bone marrow-derived dendritic cells; BSA, bovine serum albumin; CD, Crohn’s disease; CIA, collagen-induced arthritis; CII, type II collagen; CNS, central nervous system; CSF, cerebrospinal fluid; EAE, experimental autoimmune encephalomyelitis; EAT, experimental autoimmune thyroiditis; GAD65, glutamic acid decarboxylase 65; GM-CSF, granulocyte-macrophage colony-stimulating factor; IBD, inflammatory bowel disease; ID, intradermal; IFN-γ, interferon-γ; IGRP, islet-specific glucose-6-phosphatase-related protein; IM, intramuscular; HA, hemagglutinin; hmoDCs, human monocyte-derived dendritic cells; IL, interleukin; IP, intraperitoneal; IV, intravenous(ly); KO, knock-out; Kyn, kynurenine; LACK, Leishmania analog of the receptors for activated C kinase; LPS, lipopolysaccharide; mAb, monoclonal antibody; MBP, myelin basic protein; MOG, myelin oligodendrocyte glycoprotein; MRI, magnetic resonance imaging; NF-kB, nuclear factor-kappaB; NLP, nanoliposome; NMO; neuromyelitis optica; NMOSDs: neuromyelitis optica spectrum disorders; NOD, non-obese diabetic; ODN, oligodeoxynucleotide; OVA, ovalbumin; PBMCs, peripheral blood mononuclear cells; PGIA, proteoglycan-induced arthritis; PhI-p5a, Phleum pratense 5a allergen; PLGA, poly-lactic-co-glycolic acid; pMHC, major histocompatibility complex coupled with disease-relevant peptide; PS, phosphatidylserine; RA, rheumatoid arthritis; REAE, relapsing experimental autoimmune encephalitis; RR-MS, relapsing-remitting multiple sclerosis; SC, subcutaneous(ly); sc-Fv, single-chain fragment variables; SP–MS, secondary progressive multiple sclerosis; s-PS, synthetic phosphatidylserine; STAL, sialic acid-binding Ig-like lectin-engaging tolerance-inducing antigenic liposomes; T1D, type 1 diabetes; Tg, thyroglobulin; Tr1, type 1 regulatory T cell; Treg, regulatory T cell; tolDC, tolerogenic dendritic cell. * Many studies in vivo also imply assays in vitro; however, for the sake of simplicity, the in vitro tests are not directly described in the table, and the results presented in the last column may be a summary of both. ** This study is cited by Benson et al. as a personal communication as it has never been published.

## Data Availability

No new data were created or analyzed in this study. Data sharing is not applicable to this review.

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
