# Peer review of "Cutting-Edge Delivery Systems and Adjuvants in Tolerogenic Vaccines: A Review"

_pharmaceutics, 2022, doi:10.3390/pharmaceutics14091782_

Round 1

Reviewer 1 Report

This is an excellent comprehensive review of the emerging field of antigen-specific induction of immune tolerance to replace the current, less specific, clinically licenced immunosuppressants.

Minor points

Could you clarify the composition of signals 1, 2 & 3 please in Intro & table 1? I believe they refer to MHC, CD co-stimulatory molecules & cytokine microenvironment but a clearer definition would help. A graphic to emphasis this point might help?

Some text is Table 1 is illegible (e.g., panel B).

Minor revisions

Line

145 & 152            delete the “-“ before “A”

345         why “CD34+ cells” (can you explain that they’re a haematopoietic cell marker?)

490         “a DNA vaccine encoding MOG suppressed MOG-induced EAE” is confusing as it suggests its suppressive & inductive.

592         replace “contrast” with more appropriate word like “combat”

623         replace “symbolizing” with a more appropriate word

Author Response

Could you clarify the composition of signals 1, 2 & 3 please in Intro & table 1? I believe they refer to MHC, CD co-stimulatory molecules & cytokine microenvironment but a clearer definition would help. A graphic to emphasis this point might help?

The co-stimulatory cross-talk between APCs and T cells has been clarified in the introductory section dealing with immune tolerance (Paragraph 2) and in Table 1 by providing a clearer definition of co-stimulation and listing the co-stimulatory molecules and the cytokines involved in the three signals that contribute to the full activation of T cells.

Some text is Table 1 is illegible (e.g., panel B).

Figure 1 has been modified to make it more easily legible.

Line 145 & 152: delete the “-“ before “A”

Deleted

Line 345: why “CD34+ cells” (can you explain that they’re a haematopoietic cell marker?)

The reason for referring to CD34 has been explained through the sentence: “patient monocytes or their progenitors, which are recognizable for the expression of the hematopoietic cell marker CD34…..”

Line 490: “a DNA vaccine encoding MOG suppressed MOG-induced EAE” is confusing as it suggests its suppressive & inductive.

Following the Reviewer’s indication, the sentence has been modified to clarify the concept and avoid confusion during interpretation.

Line 592: replace “contrast” with more appropriate word like “combat”

Replaced with “combat”.

Line 623: replace “symbolizing” with a more appropriate word

Replaced with “a marker of apoptosis”.

Reviewer 2 Report

In this review, Puricelli et al. summarizes the different natural regulatory steps leading to immunotolerance offering various levels and key mechanisms to tackle in order to induce tolerance, in particular using tolerogenic vaccines.

The author provides a review of different tolerogenic vaccine strategies, including vaccines aiming to increase anergy, providing vaccine platform delivering Ag without costimulatory molecules or co-delivering molecules reducing inflammatory signals; or to increase natural physiological tolerance (via oral delivery for instance or additional mechanisms exploiting natural pathways such as Dendritic cell-based strategies and nucleic acid-based vaccines).

The review is clear, interesting, well-constructed and well documented.

I only have specific comments as follow:

·      The Table 1 appears to include a repetition of the end part of the table: of the section ‘ Metabolic induction of regulatory T cells’. Although it could result from an artefact on the peer review version of the review, should it still be present in the last version, it should be addressed.

·      The abbreviations ‘AHR’ (aryl hydrocarbon receptor (AHR) should be included in the abbreviations section below table 1.

·      I would suggest to include one sentence to describe in more details the AHR- Kyn pathway, that is later introduced in the Table 1, in the corresponding paragraph line 160-164.

·      Similarly, I would suggest to introduce in more details the RALD2-RA pathway, earlier in the text: line 164-167 as it is later described (line 241-244).

Author Response

The Table 1 appears to include a repetition of the end part of the table: of the section ‘ Metabolic induction of regulatory T cells’. Although it could result from an artefact on the peer review version of the review, should it still be present in the last version, it should be addressed.

The authors are grateful for the notification of this redundancy. The repeated part of Table 1 has been deleted.

The abbreviations ‘AHR’ (aryl hydrocarbon receptor (AHR) should be included in the abbreviations section below table 1.

The abbreviation has been included. Thank you for noticing this inattention.

I would suggest to include one sentence to describe in more details the AHR- Kyn pathway, that is later introduced in the Table 1, in the corresponding paragraph line 160-164.

The AhR-Kyn pathway has been clarified with more details about the intracellular signaling pathways and the consequances of Kyn-AhR interaction on the immune response.

Similarly, I would suggest to introduce in more details the RALD2-RA pathway, earlier in the text: line 164-167 as it is later described (line 241-244).

A short paragraph on the key role of retinoic acid in tolerance induction has been added in the section suggested by the Reviewer to introduce the concept before describing it later in 

Reviewer 3 Report

 1. Please improve introduction and conclusion.

2. References need to be past 5 years unless important.

3. Manuscript need table on its application in in vitro, and in vivo animal model.

4. Authors should add other figures for other section in the manuscript. It is necessary.

5. The clinical applications and studies are missing.

6. The English needs to be checked and corrected by a native English writer.

Author Response

Please improve introduction and conclusion.

 The introductory section has been revised and some short paragraphs have been added especially to the general description of immune tolerance to provide more details about concepts that had been only briefly mentioned (such as the kynurenine and retinoic acid pathways). Also following the Reviewer’s indication n. 5 about the lack of clinical applications, the conclusion has been broadened with a paragraph on the clinical fields where tolerogenic vaccines might represent a promising therapeutic strategy.

References need to be past 5 years unless important.

Many studies, especially the ones on animal models, were performed more than 5 years ago. The Authors believe they are worth citing to support evidence for what is described in this review.

Manuscript need table on its application in in vitro, and in vivo animal model.

A second table (Table 2) has been added with a list of the preclinical and clinical studies cited in the review with a description of their main characteristics and clinical applications.

Authors should add other figures for other section in the manuscript. It is necessary.

A second figure has been added to clarify the phenomenon of epitope spreading, thinking that its complexity might be better understood through a graphical explanation.

The clinical applications and studies are missing.

 A paragraph has been added to the “Concluding remarks” section to better define the clinical applications of tolerogenic vaccines, including some references to clinical trials performed on human subjects.

The English needs to be checked and corrected by a native English writer.

The authors are willing to submit the manuscript to a native English writer for language revision. However, his contribution will be available only from 29th August owing to the holiday period, which typically occurs during the last two weeks of August in Italy. If a submission deadline extension were possible, the Authors would send him the manuscript as soon as he is available.

Round 2

Reviewer 3 Report

All corrections are well done and accept in present form.